# Biosynthesis of cannabinoid precursor olivetolic acid in genetically engineered *Yarrowia lipolytica*

Jingbo Ma[1,2], Yang Gu[1,3] & Peng Xu [1,4 ✉]

Engineering microbes to produce plant-derived natural products provides an alternate solution to obtain bioactive products. Here we report a systematic approach to sequentially identify the rate-limiting steps and improve the biosynthesis of the cannabinoid precursor olivetolic acid (OLA) in *Yarrowia lipolytica*. We find that *Pseudomonas sp* LvaE encoding a short-chain acyl-CoA synthetase can efficiently convert hexanoic acid to hexanoyl-CoA. The co-expression of the acetyl-CoA carboxylase, the pyruvate dehydrogenase bypass, the NADPH-generating malic enzyme, as well as the activation of peroxisomal β-oxidation pathway and ATP export pathway are effective strategies to redirect carbon flux toward OLA synthesis. Implementation of these strategies led to an 83-fold increase in OLA titer, reaching 9.18 mg/L of OLA in shake flask culture. This work may serve as a baseline for engineering cannabinoids biosynthesis in oleaginous yeast species.

[1] Department of Chemical, Biochemical and Environmental Engineering, University of Maryland, Baltimore County, Baltimore, MD 21250, USA. [2] Present address: College of Biological and Pharmaceutical Engineering, West Anhui University, Lu'an, Anhui 237012, China. [3] Present address: School of Food Science and Pharmaceutical Engineering, Nanjing Normal University, Nanjing 210023, China. [4] Present address: Department of Chemical Engineering, Guangdong Provincial Key Laboratory of Materials and Technologies for Energy Conversion (MATEC), Guangdong Technion-Israel Institute of Technology, Shantou, Guangdong 515063, China. ✉email: peng.xu@gtiit.edu.cn

annabinoids are a large family of neurological active compounds. Cell biologists have identified multiple cannabinoids receptors in human brain and found cannabinoids or their derivatives played an important role to regulate human cognitive and emotional functions[1,2]. Recent pharmaceutical and clinical studies have proved that cannabinoids can be used to treat anxiety, depression, aging-related muscle/joint pain, seizure, stroke and cardiovascular diseases etc[3,4]. Almost all cannabinoids are sourced from farm-harvested hemp or *Cannabis sativa* (*C. sativa*)-related species. Recent molecular breeding has resulted in special hemp cultivars which synthesize less tetrahydrocannabinol (THC) -- the principle psychoactive constituent of cannabis (*Marijuana*)[5]. Cannabidiol (CBD) oils free of THCs will not cause human addiction. FDA and World Health Organization (WHO) have legally approved the use of CBDs as an essential drug ingredient to treat neurological disorders. Recently, a growing number of nations have lifted the restrictions and approved the use of THC-free CBDs as a consumer chemical in human food, drink, cosmetics and nutraceutical industry. For example, hemp-derived CBD oil (free of THC) has been infused with human lotion, confectionary (gummies or cookies) or cigarette products. The overall market size of CBD oil as consumer chemicals is estimated to be $25 billion in 2025 (assuming 50 Million people will use CBDs as a consumer chemical daily, average dosage 50 g/people/year or 2 mg/kg-body-weight/day). Current crude CBD oil (45% strength) is sold at about $3000/kg and pure CBD oil is sold at $10,000/kg. CBD market is projected to increase 10–15% annually, the current hemp planting and agricultural technology cannot meet this rapid market demand. Covid-19 has severely disrupted our supply chain of food, pharmaceuticals and consumer chemicals, driving us to seek alternate solutions to manufacture essential drugs and safeguard our well-being[6]. There is an urgent need to develop alternate route to fill the CBD supply chain.

Microbial metabolic engineering is considered as the enabling technology to mitigate environmental concerns and address the scarcity of resource limitation challenges[7,8]. Compared to plant-based production system, microbes have a number of advantages, including less dependence on arable land or climate changes, ease of genetic manipulation and large-scale production, as well as robust growth and high conversion rate across a wide range of low-cost renewable raw materials[6,9]. Therefore, genetically modified microbes have been widely used to produce fuels, commodity chemicals and nutraceuticals[10]. With our increased knowledge of cellular physiology and understanding of molecular genetics toward higher and complex organisms, there is a growing interest to develop novel cellular chassis that may overcome the constraints of common hosts (*Escherichia coli* and *Saccharomyces cerevisiae*)[11]. For example, *Yarrowia lipolytica*, is characterized as a generally regarded as safe (GRAS) oleaginous yeast, has been recently modified to produce an arrange of value-added natural products, including resveratrol[12], squalene[13], flavonoids[14], artemisinin[15] and violacein[16] etc. The distinct oil-accumulating (hydrophobic) environment, abundance of internal membrane structure, and the compartmentalization of biosynthetic pathways in *Y. lipolytica* provide the ideal microenvironment for the catalytic functions of enzymes with stereo- or regio-selectivity[11]. This property is extremely important for the functional expression of P450 enzymes with site-specific hydroxylation or peroxidation reactions. CBD biosynthetic pathway involves polyketide synthase, mevalonate pathway and prenyltransferase, which posts a considerable challenge for efficient biosynthesis in common host organism[17]. As an oleaginous yeast, *Y. lipolytica* is reported to accommodate strong flux for acetyl-CoA, malonyl-CoA and HMG-CoA. Naturally, *Y. lipolytica* could be a promising host to produce CBDs and their derivatives. In this work, we explored the possibility to use oleaginous yeast *Y. lipoltyica* as the host to synthesize olivetolic acid, a universal precursor to synthesize CBDs. We overcome a number of critical pathway bottlenecks to unlock the potential of *Y. lipolytica* to synthesize olivetolic acid. These strategies, when combined, lead to a more than 83-fold increase in olivetolic acid production. This preliminary result may provide a baseline for us to develop CBD-producing oleaginous yeast cell factories.

## Results and discussion

**Biosynthesis of olivetolic acid in engineered *Y. lipolytica*.** Olivetolic acid (OLA) biosynthesis involves the condensation of one hexanoyl-CoA with three malonyl-CoAs by a type III PKS (tetraketide synthase) OLA synthase (*C. sativa* OLS; *Cs*OLS) and an OLA cyclase (*Cs*OAC) (Fig. 1), both enzymes were derived from the *Cannabis* species (*C. sativa*). Previous studies have achieved trace amount of OLA production in both *E. coli* and *S. cerevisiae*[17,18]. To produce OLA in *Y. lipolytica*, the plasmid pYLXP'-*CsOLS*-*CsOAC* encoding the codon-optimized *CsOLS* and *CsOAC* was transformed into *Y. lipolytica*. Under HPLC characterization, OLA has a characteristic retention time of 10.8 min (Fig. 2a). The resulting strain YL101 only produced 0.11 mg/L OLA after 96 h cultivation (Fig. 2b), which is comparable to the initial OLA production in *S. cerevisiae*[17].

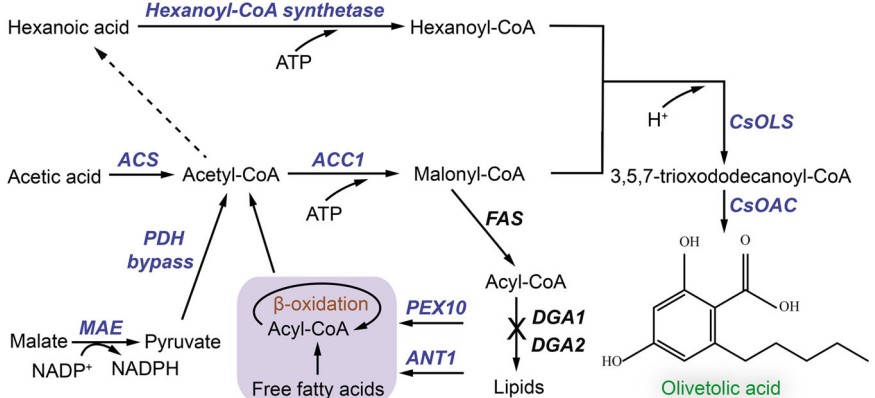

**Fig. 1 Metabolic engineering strategies to improve olivetolic acid (OLA) synthesis in *Y. lipolytica*.** ACC1 acetyl-CoA-carboxylase, ACS acetyl-CoA synthase, ANT1 adenine nucleotide transporter, *Cs*OAC OLA cyclase from *Cannabis sativa*, *Cs*OLS, OLA synthase from *C. sativa*, DGA1 and DGA2 diacylglycerol acyltransferases, FAS fatty acid synthase, MAE malic enzyme, PDH pyruvate dehydrogenase, PEX10 peroxisomal matrix protein. Blue colored steps are beneficial for OLA production.

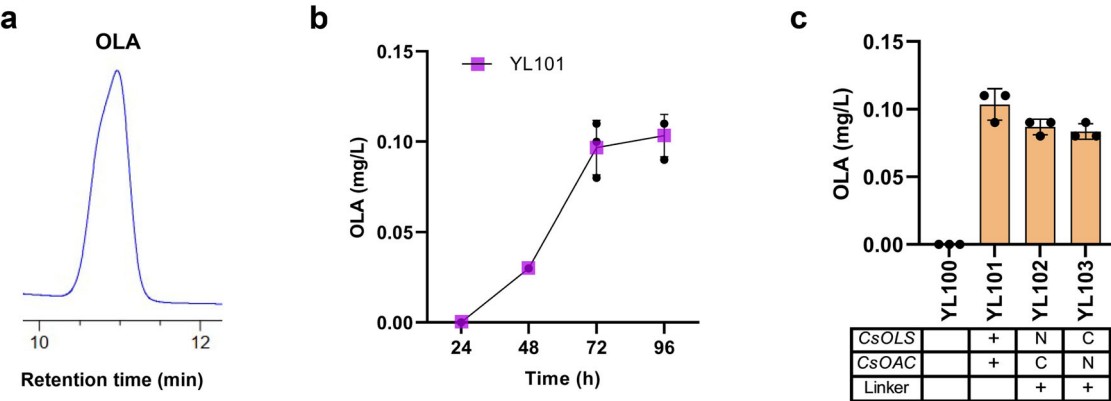

**Fig. 2 Production of olivetolic acid (OLA) by configuring OLS and OAC. a** In vivo production of OLA in YL101 strain. Extracts were analyzed by LC-MS and signals were compared to authentic OLA standards (supplementary files). **b** Time profiles of OLA titer in YL101 strain. **c** OLA production in engineered strains by expressing OLS and OAC or their fusions. Error bars represent standard deviation of three biological replicates ($n = 3$).

Evidence showed that some undesired byproducts could also be formed during OLA accumulation[17,18]. We hypothesized that fusing *CsOLS* with *CsOAC* may place the two enzymes in close proximity and minimize the dissipation of intermediates. In such a way, fused OLS-OAC may efficiently transfer the intermediates to form OLA with minimal byproducts. We fused the two proteins with an amino acid linker (10×Glycine) in two different orientations. When *CsOLS* was fused to the N-terminus or C-terminus of the *CsOAC*, the obtained strains YL102 and YL103 showed a slightly decline in the production of OLA compared with the control strains YL101 (Fig. 2c). Possibly, the fusion of these two proteins negatively impacted protein folding resulting in suboptimal catalytic functions. Since *CsOLS-CsOAC* fusion could not further improve OLA, the best production strain, YL101, was subjected to further engineering.

**Screening enzymes to improve pool size of hexanoyl-CoA.** Pool size of hexanoyl-CoA was proven to be a rate-limiting factor in both *E. coli* and *S. cerevisiae*[17,18]. An acyl activating enzyme encoded by *CsAAE1* from *Cannabis sativa* was characterized to catalyze the formation of hexanoyl-CoA from hexanoic acid, and the expression of *CsAAE1* has been shown to successfully increase the titer of OLA by twofold in *S. cerevisiae*[17]. Hexanoic acid is toxic and the dosage of hexanoic acid (HA) should be optimized to minimize its negative impact on cell fitness[18]. Thus, *CsAAE1* was co-expressed with *CsOLS* and *CsOAC* in *Y. lipolytica* supplemented with 0.5 mM, 1 mM, and 2 mM of HA. We found that OLA production was declined (Fig. 3a). The yeast grows slowly with HA supplementation, confirming the negative effect of HA on cell growth (Fig. 3b). To mitigate this effect, we delayed HA supplementation to 48 h. OLA production with 0.5 mM HA supplementation slightly increased to 0.13 mg/L (Fig. 3a), and the yeast growth fitness remains unchanged (Fig. 3c). Thus, the supplementation of 0.5 mM hexanoic acid at 48 h was used for further strain screening.

Nevertheless, codon-optimized *CsAAE1* showed lower catalytic capacity in *Y. lipolytica* than in *S. cerevisiae*[17]. In order to optimize the conversion of HA to hexanoyl-CoA, we screened a panel of six enzymes that have been annotated with acyl-CoA synthetase activity. *CsAAE3* encoding a peroxisomal enzyme can accept a variety of fatty acid as substrates including HA. We also removed the C-terminal peroxisome targeting sequence 1 (PTS1) and overexpressed the codon-optimized *CsAAE3* in *Y. lipolytica*[19–21]. Both the short-chain and long-chain fatty acyl-CoA synthetases, *Ec*FadK and *Ec*FadD from *E. coli*, have been shown to exhibit a catalytic activity on C6–C8 fatty acids[18].

The medium-chain fatty acyl-CoA synthetase *Sc*FAA2 from *S. cerevisiae* and the *Y. lipolytica* native fatty acyl-CoA synthetase encoded by *ylFAA1* (YALI0D17864g) were also tested in this study. *Pseudomona sp*, known as a superior environmental degrader for various carbon sources. The *LvaE* gene encoding the short-chain acyl-CoA synthetase of *Pseudomonas putida* KT2440 (*PpLvaE*, PP_2795 with uniport ID Q88J54), which is reported to active with C4-C6 carboxylic acids, was also selected[22]. Among these chosen hexanoyl-CoA synthetases, we found the over-expression of *PpLvaE* with *CsOLS* and *CsOAC* (strain YL110) resulted in the highest OLA production, an eightfold increase in OLA titer (1.07 mg/L) (Fig. 4a). *P. putida* has been reported to exhibit better solvent tolerance and grow under a number of harsh or hydrophobic conditions. The overexpression of *CsAAE1* in *S. cerevisiae* only showed a twofold increase in OLA titer[17]. By contrast, *PpLvaE* outperforms other candidate enzymes in converting HA to hexanoyl-CoA. To the best of our knowledge, this is the first report that harnessed *PpLvaE* to overcome a rate-limiting step in OLA and cannabinoids pathway. Future enzymology study or directed evolution of *PpLvaE* may improve its catalytic function. Apparently hexanoic acid toxicity poses severe inhibition to cell growth, and the hexanoyl-CoA ligase needs to be further improved. An evolutionary approach to improve hexanoic acid tolerance and selection of variants of hexanoyl-CoA ligase will be important to solve this challenge. Alternative approach is to harness the endogenous β-oxidation pathway and target OA pathway into the peroxisome.

**Boosting malonyl-CoA to improve olivetolic acid production.** Malonyl-CoA has been reported as the rate-limiting precursors for polyketides synthesis[23]. Next, we used two strategies to improve malonyl-CoA supply. In the first strategy, two endogenous genes *ylDGA1* (YALI0E32769g) and *ylDGA2* (YALI0D07986g) encoding diacylglycerol acyltransferases were knocked out, in such way, we could minimize the acyl-CoA flux flowing into the TAG pathway which competes with OA pathway for malonyl-CoA[24,25]. When the OA pathway (*CsOLS-CsOAC-PpLvaE*) was transformed into the strain YL111 with *ylDGA2* knockout and YL112 strain with *ylDGA1* and *ylDGA2* double knockout, the resulting strains YL113 and YL114 showed a dramatic decline in OLA production (Fig. 4b). We speculate that that the increased acyl-CoAs, or acyl-ACPs in the DGA1/DGA2 knockout strains may strongly feedback inhibit the activity of the acetyl-CoA carboxylase[26,27], which has been reported in both prokaryotic and eukaryotic cells.

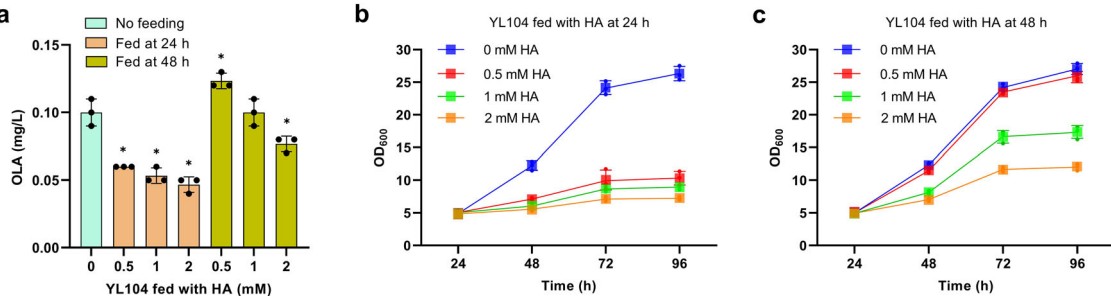

**Fig. 3 Olivetolic acid (OLA) and cell growth profile of YL104 strain expressing *CsAAE1*,*CsOLS* and *CsOAC* with the supplementation of hexanoic acid (HA). a** OLA titers in YL104 strain supplied with 0.5 mM, 1 mM, and 2 mM hexanoic acid at 24 h or 48 h. **b** Cell growth profile of YL104 strain fed with different dosages of HA at 24 h. **c** Cell growth profile of YL104 strain fed with different dosages of HA at 48 h. Error bars represent standard deviation of three biological replicates ($n = 3$). The asterisk indicates the statistical significance at a $p < 0.05$ (two-tailed *t*-test).

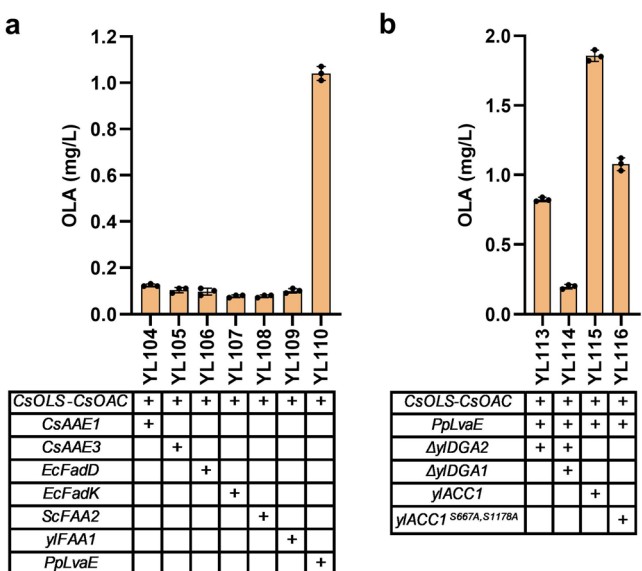

**Fig. 4 Boosting hexanoyl-CoA and malonyl-CoA to improve olivetolic acid (OLA) production. a** OLA titers in engineered strains by expressing different hexanoyl-CoA synthetases for optimizing the conversion of hexanoic acid to hexanoyl-CoA. **b** OLA titers in engineered strains for enhancing the availability of malonyl-CoA. Error bars represent standard deviation of three biological replicates ($n = 3$).

In the second strategy, the endogenous gene *ylACC1* (YALI0C11407g) encoding acetyl-CoA carboxylase was over-expressed to facilitate the conversion of acetyl-CoA to malonyl-CoA[28]. To minimize spliceosomal or transcriptional level regulation, the two internal introns of the native gene *ylACC1* were removed. The engineered strain YL115 with overexpression of yl*ACC1* produced 1.9 mg/L of OLA, with a 1.8-fold improvement (Fig. 4b). Additionally, snf1 (AMP kinase) has been reported to phosphorylate and inhibit ACC activity[29,30]. Recently, a snf1-resistant version of ACC mutant (*ScACC1*[S659A,S1157A]) has been reported to boost intracellular malonyl-CoA and improve 3-hydroxyl propionate or fatty acids production in *S. cerevisiae*[31]. Thus, a SNF1-resistant version of *ylACC1*[S667A,S1178A] was engineered and incorporated into strain YL116. In *Y. lipolytica*, we did find the SNF1 homolog protein (YALI0D02101). However, OLA production in strain YL116 failed to improve further (Fig. 4b), possibly due to some hidden or underexplored regulations in *Y. lipolytica*.

**Effect of pH on olivetolic acid production**. During fermentation, we found that the culture pH dramatically dropped to below 3.5. This strong acidity may negatively affect membrane permeability and strain performance[28,32]. We next sought to control the medium pH by using either PBS buffer or CaCO₃[32]. Supplementation of 20 g/L CaCO₃ maintained stable pH and increased OLA titer by threefold, reaching 5.86 mg/L at 96 h, whereas PBS failed to improve OLA production (Fig. 5a). We speculate that PBS buffer may shift the cell metabolism or change membrane permeability[33], which negatively impact cell performance.

**Debottlenecking acetyl-CoA, ATP and NADPH supply to improve olivetolic acid production**. Acetyl-CoA served as the basic building block for both hexanoyl-CoA and malonyl-CoA, and the biosynthesis of OLA from acetyl-CoA requires extensive consumption of ATP and NADPH as cofactors to facilitate the activity of ACC, hexanoyl-CoA synthetase and the tetraketide synthase[17,18,34]. To simplify genetic manipulations and increase strain stability, we next integrated the gene cassettes containing *CsOLS-CsOAC-PpLvaE-ylACC1* at the genomic locus of YALI0C05907g, which was screened as an orthogonal integration site for polyunsaturated fatty acid production in *Y. lipolytica*[35]. Each of the gene was placed under control of an independent TEF-intron promoter and XPR2 terminator. The integrated strain YL117 yielded about 3.26 mg/L OLA in YPD complex media, which was lower than the production level (5.86 mg/L) of the strain YL115 with chemically-defined complete synthetic media (CSM-leu), possibly due to the altered gene expression profile resulting from genomic integration or shifting of culture media from CSM to YPD (Fig. 5b). To probe the effect, YL117 strain was restored the leucine marker by transformation of the empty pYLXP' vector and fermented in CSM-leu media, but the OLA titer decreased to 2.99 mg/L, which indicates YPD media was better for the integrated strain (Fig. 5b). Subsequently, the genomic locus of *ku70* and the pBR docking site[16] were chosen as integration site for comparison with the genomic locus of YALI0C05907g, resulting in the integrated strains YL118 and YL119, respectively. The strain YL118 in YPD media produced a comparatively higher OLA titer of 3.54 mg/L (Fig. 5b). By comparisons of the OLA titer from the integrated strains, we speculated that the supply of acetyl-CoA may create a major bottleneck for OLA synthesis in *Y. lipolytica*.

*Y. lipolytica* peroxisomal matrix protein Pex10 (*yl*Pex10)[36], *Salmonella enterica* acetyl-CoA synthetase mutant (*Se*Acs[L641P])[37], and *E. coli* pyruvate dehydrogenase complex (*Ec*PDH) with the lipoate-protein ligase A (*Ec*LplA)[28] were reported to boost the level of acetyl-CoA in *Y. lipolytica* or *S. cerevisiae*. Thus, we overexpressed these enzymes and their combination in the strain YL118. The resultant strains improved the OLA production to different extents, with the strain YL125 overexpressing *yl*Pex10,

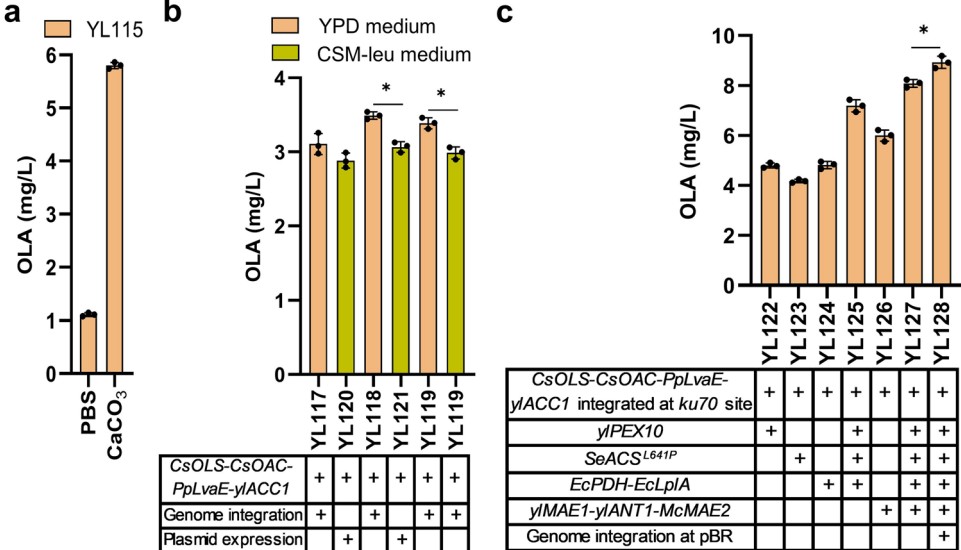

**Fig. 5 Improve olivetolic acid (OLA) production by controlling pH and increasing acetyl-CoA, ATP and NADPH supply. a** OLA titers after controlling pH of the fermentation medium by using either PBS buffer or CaCO₃. **b** OLA titers in strains with relevant genes integrated into different genomic loci cultivated in YPD or CSM-leu media. **c** OLA titers in engineered strains by expressing different enzymes and their combinations for improving the supply of acetyl-CoA, ATP and NADPH. Error bars represent standard deviation of three biological replicates ($n = 3$). The asterisk indicates the statistical significance at a $p < 0.05$ (two-tailed $t$-test).

$SeAcs^{L641P}$ and $Ec$PDH yielding the highest OLA titer of 7.43 mg/L (Fig. 5c). Ant1p, a peroxisomal adenine nucleotide transporter, is an integral protein of the peroxisomal membrane and is responsible for transferring ATP into peroxisomes for β-oxidation of medium-chain fatty acids, which will increase the rate of β-oxidation cycle and provide cytosolic acetyl-CoAs[38]. For example, the $Sc$ANT1 has been overexpressed in *S. cerevisiae* to supply ATP and enhance peroxisome functions[39]. Thus, we chose to overexpress the $Sc$ANT1 homologous gene $yl$ANT1 (YALI0E03058g) encoding *Y. lipolytica* peroxisomal ATP/AMP transporter to increase ATP supply and the peroxisomal β-oxidation rate. The native gene $yl$MAE1 (YALI0E18634g) encoding malic enzyme (MAE) and $Mc$MAE2 from *Mucor circinelloides* were reported to provide NADPH in *Y. lipolytica*[28,40]. In addition, pyruvate can be generated from malic enzyme, and the resulting pyruvate can be converted to acetyl-CoA by $Ec$PDH with $Ec$LplA, which synergistically boosted cytosolic acetyl-CoA in *Y. lipolytica*[32]. We next transformed the plasmid pYLXP'-$yl$MAE1-$yl$ANT1-$Mc$MAE2 into the strain YL118 to improve the supply for both ATP and NADPH. The resultant strain YL126 produced 6.21 mg/L of OLA (Fig. 5c). Next, the expression plasmid containing seven genes which overcome precursor limitations for acetyl-CoA, NADPHs and ATPs were constructed and then transformed into YL118 for co-expression. The resulting strain YL127 further enhanced OLA production to 8.23 mg/L (Fig. 5c). Finally, the linearized gene fragments containing pYLXP'-$yl$PEX10-$SeACS^{L641P}$-$Ec$PDH-$Ec$L-plA-$yl$MAE1-$yl$ANT1-$Mc$MAE2 was integrated at the pBR docking site of the strain YL118. The engineered strain YL128 cultured in YPD media produced OLA with a titer of 9.18 mg/L (Fig. 5c), which was improved by 83 times as compared to the OLA titer produced by the initial strain YL101 (0.11 mg/L). The current OLA production titer represents a threefold higher than the reported OLA titer produced in engineered *S. cerevisiae* in the shaker flasks[17].

## Conclusions

In this study, we have systematically investigated the enzymatic bottlenecks that restrain the efficient biosynthesis of cannabinoid precursor olivetolic acid in the oleaginous yeast *Y. lipolytica*. We took a reverse engineering approach and sequentially identified that the supply of hexanoyl-CoA, malonyl-CoA, acetyl-CoA, NADPH and ATP are the rate-limiting steps. To overcome these limitations, we have screened enzymes aiming to debottleneck the pathway limitations and redirect the carbon flux toward the end-product olivetolic acid. We discovered that *P. putida* LvaE encoding an acyl-CoA synthetase could efficiently convert exogenously-added hexanoic acid to hexanoyl-CoA. The co-expression of the acetyl-CoA carboxylase, the pyruvate dehydrogenase bypass, the NADPH-generating malic enzyme, as well as the activation of peroxisomal β-oxidation pathway and ATP export pathway were efficient strategies to remove the pathway bottlenecks. Collectively, these strategies have led us to construct a *Y. lipolytica* strain that produced olivetolic acid at a titer 83-fold higher (9.18 mg/L) than our initial strain (0.11 mg/L). While the current production level is low, we expect these strategies may serve as a baseline for other metabolic engineers who are interested in engineering cannabinoid biosynthesis in oleaginous yeast species.

## Methods

**Strains, plasmids, primers, and chemicals.** All strains of engineered *Y. lipolytica*, including the genotypes, recombinant plasmids, and primers have been listed in Supplementary Table 1, Supplementary Table 2 and Supplementary Table 3. Olivetolic acid was purchased from Santa Cruz Biotechnology. All other chemicals were obtained from Sigma-Aldrich and Fisher Scientific. Heterologous synthetic genes including genes *CsOLS*, *CsOAC*, *CsAAE1*, *CsAAE3*, *PpLvaE*, $SeACS^{L641P}$ and *McMAE2* were codon-optimized using the online IDT Codon Optimization Tool and then ordered from GENEWIZ (Suzhou, China). The codon-optimized gene sequence could be found in Supplementary Note 1. The synthetic gene fragments were assembled with the New England Biolab Gibson Assembly kits, with the pYaliBrick vector as plasmid backbone. DNA sequences were verified by Sanger sequencing (QuintaraBio).

**Shake flask cultivations and pH control.** For performing shake flask cultivations, seed culture was carried out in the shaking tube with 2 mL seed culture medium at 30 °C and 250 r.p.m. for 48 h. Then, 0.6 mL of seed culture was inoculated into the 250 mL flask containing 30 mL of fermentation medium and grown under the conditions of 30 °C and 250 r.p.m. for 96 h. One milliliter of cell suspension was sampled every 24 h for $OD_{600}$ and desired metabolite measurement.

Seed culture medium used in this study included the yeast complete synthetic media regular media (CSM, containing glucose 20.0 g/L, yeast nitrogen base without ammonium sulfate 1.7 g/L, ammonium sulfate 5.0 g/L, and CSM-Leu 0.74 g/L) and complex medium (YPD, containing glucose 20.0 g/L, yeast extract 10.0 g/L, and peptone 20.0 g/L). Fermentation medium used in this study contained the yeast complete synthetic media regular media (CSM, containing glucose 40.0 g/L, yeast nitrogen base without ammonium sulfate 1.7 g/L, ammonium sulfate 1.1 g/L, and CSM-Leu 0.74 g/L) and complex medium (YPD, containing glucose 40.0 g/L, yeast extract 10.0 g/L, and peptone 20.0 g/L). To control the pH, 20 mM phosphate buffer saline (PBS, $Na_2HPO_4$–$NaH_2PO_4$) or 20 g/L $CaCO_3$ was used, respectively.

**Yeast transformation and screening of high-producing strains**. The standard protocols of *Y. lipolytica* transformation by the lithium acetate method were described as previously reported[16,41]. In brief, one milliliter cell was harvested during the exponential growth phase (16–24 h) from 2 mL YPD medium (yeast extract 10 g/L, peptone 20 g/L, and glucose 20 g/L) in the 14-mL shake tube, and washed twice with sterile 100 mM phosphate buffer (pH 7.0). Freshly cultivated yeast colony lawns picked from overnight-grown YPD plates could also be used for genetic transformation. Then, cells were resuspended in 105 μL transformation solution, containing 90 μL 50% PEG4000, 5 μL lithium acetate (2 M), 5 μL boiled single stand DNA (salmon sperm, denatured) and 5 μL DNA products (including 200–500 ng of plasmids, linearized plasmids or DNA fragments), and incubated at 39 °C for 1 h, then spread on selected plates. It should be noted that the transformation mixtures needed to be vortexed for 15 s every 15 min during the process of 39 °C incubation. The selected markers, including leucine, uracil and hygromycin, were used in this study. All engineered strains after genetic transformation were undergone PCR screening using the GoTaq Green PCR kits, and the strain with the correct gene fragment was selected to perform shake flask cultivation. For shaking tube cultivations, 100 μL seed cultures were inoculated into 5 mL fermentation media in a 50 mL tube.

**Single-gene and multi-genes expression vectors construction**. In this work, the YaliBrick plasmid pYLXP' was used as the expression vector[42]. The process of plasmid constructions followed the YaliBrick gene assembly platforms[16]. In brief, recombinant plasmids of pYLXP'-xx (a single gene) were built by Gibson Assembly of linearized pYLXP' (digested by *SnaBI* and *KpnI*) and the appropriate PCR-amplified or synthetic DNA fragments. Multi-genes expression plasmids were constructed based on restriction enzyme subcloning with the isocaudamers *AvrII* and *NheI*. All genes were respectively expressed by the TEF promoter with intron sequence and XPR2 terminator, and the modified DNA fragments and plasmids were sequence-verified by Sanger sequencing (Quintarabio).

**Gene knockout**. A marker-free gene knockout method based on Cre-*lox* recombination system was used as previously reported[16,43]. For performing gene knockout, the upstream and downstream sequences (both 1000 bp) flanking the deletion targets were PCR-amplified. These two fragments, the *loxP-Ura/Hyr-loxP* cassette (digested from plasmid pYLXP'-*loxP-Ura/Hyr* by *AvrII* and *salI*), and the gel-purified plasmid backbone of pYLXP'(linearized by *AvrII* and *salI*) were joined by Gibson Assembly, giving the knockout plasmids pYLXP'-*loxP-Ura/Hyr-xx* (xx is the deletion target). Next, the knockout plasmids were sequence-verified by Quintarabio. Then, the gene knockout cassettes were PCR-amplified from the knockout plasmids pYLXP'-*loxP-Ura/Hyr-xx*, and further transformed into *Y. lipolytica*. The positive transformants were determined by colony PCR. Knockout strains were built on top of the Ku70-deficient strains (Po1f background). Subsequently, plasmid pYLXP'-*Cre* was introduced into the positive transformants and promoted the recombination of *loxP* sites, which recycle the selected marker. Finally, the intracellular plasmid pYLXP'-*Cre* was evicted by incubation at 30ºC in YPD media for 48 h. Here, *Ura* is the uracil marker, and *Hyr* is hygromycin marker.

**Genomic integration of desired genes**. In this work, genomic integration of desired genes was performed in two different ways: site-specific genomic integration plasmids or application of pBR docking platform by linearizing the plasmid pYLXP' with digested enzyme *NotI*. We constructed two genomic integration plasmids pURLA and pURLB, corresponding to the *Ku70* and *YALI0C05907g* (encoding a hypothetical protein conserved in the *Yarrowia* clade) genomic sites, respectively. The procedure of using these two plasmids was similar as that of gene knockout protocol. The method of constructing integration plasmids was described in previous work. The application of pBR docking platform was achieved by linearizing the plasmid pYLXP' with *NotI* restriction enzyme digestion. All the genes manipulated in this study are one copy.

**HPLC quantification and LC–MS characterization of olivetolic acid**. Cell densities were monitored by measuring the optical density at 600 nm ($OD_{600}$). The concentrations of olivetolic acid were measured by high-performance liquid chromatography (HPLC) through Agilent HPLC 1220. In detail, olivetolic acid was measured at 270 nm under 40 °C (column oven temperature) with a mobile phase containing 60% (v/v) methanol in water at a flow rate of 0.4 mL/min equipped with a ZORBAX Eclipse Plus C18 column (4.6 × 100 mm, 3.5 μm, Agilent) and the VWD detector.

To quantify the concentration of olivetolic acid, 0.5 mL whole cell sample with both cell pellet and liquid culture was taken. Subsequently, samples were treated with 2 U/OD zymolyase (2 h, 30 °C with shaking at 1000 r.p.m.), and then cell suspensions were added with 20% (w/v) glass beads (0.5 mm) and the cells were grinded with hand-powered electrical motor (VWR). Subsequently, the crude extracts were mixed with an equal volume of ethyl acetate (v/v), followed by vortex at room temperature for 2 h. Organic and inorganic layers were separated by centrifugation at 12,000 r.p.m. for 10 min. Samples were extracted three times. The combined organic layers were evaporated in a vacuum oven (50 °C) and the remainders were resuspended in 0.5 mL 100% methanol. Then, 100 μL of the sample were gently transferred into a HPLC vial insert and 5 μL were injected into HPLC for OLA quantification. Under this condition, the retention time for OLA is 10.8 min.

Olivetolic acids standards and samples were also characterized by the Perkin Elmer QSight LX50 UHPLC (ultra-high-performance liquid chromatography) with the QSight 210 Mass Spectrometer under negative ESI (electrospray ionization) mode. The Agilent Zorbax Eclipse XDB-C18 2.1 × 50 mm, 1.8 μm (P/N 981757-902) column was used for the UHPLC system. Under the UHPLC-MS system, the retention time for olivetolic acid is 1.8 min and the characteristic mass/charge ($m/z$) ratio for OLA is 223.5 and 179.5 (decarboxylated-$CO_2$ fragment). A detailed UHPLC-MS report for OLA could be found in the Supplementary note 2.

**Statistics and reproducibility**. General data analysis (means and standard deviation) was performed primarily by GraphPad Prism 9.4.0. All experiments were performed with biological triplicates and values were expressed as means ± standard errors.

**Reporting summary**. Further information on research design is available in the Nature Portfolio Reporting Summary linked to this article.

## Data availability

All data in this published article can be found in Supplementary Data file.

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

## Acknowledgements

This project is supported by the Bill & Melinda Gates Foundation Award (OPP1188443) and National Science Foundation (CBET-1805139). We thank Dr. Joshua Wilhide and Dr. LaCourse to help run the LC–MS characterization of OLA at the UMBC Molecular Characterization and Analysis Complex center (UMBC chemistry facility) and help prepare the LC–MS report.

## Author contributions

P.X. designed the study, analyzed the data and revised the manuscript. J.M. performed the genetic engineering, enzyme screening and cell cultivation with input from Y.G.. J.M. wrote the manuscript and analyzed the data. J.M. and P.X. revised the manuscript.

## Competing interests

The authors declare no competing interests.
