## [Peer Review File · Communications Biology]

Reviewers' comments:

Reviewer #1 (Remarks to the Author):

In this manuscript, the authors engineer an oleaginous yeast, *Yarrowia lipolytica*, to produce olivetolic acid, a key precursor of cannabinoid biosynthesis. The authors optimize the metabolic pathway, especially the biosynthesis of Hexanoyl-CoA and Malonyl-CoA. Although previous studies have successfully constructed *E. coli* and *S. cerevisiae* strains to produce olivetolic acid and cannabinoids, the authors argue that *Yarrowia* could be a promising host in large-scale cannabinoid production.

My major concern is that the high toxicity of hexanoic acid to *Yarrowia* would be the bottleneck of olivetolic acid production. Have the authors tried to synthesize Hexanoyl-CoA from Acetyl-CoA as Ref. 22 did?

Previous studies, such as Ref. 22, had successfully produced olivetolic acid by overexpressing CsAAE1, CsOLS and CsOAC while feeding hexanoic acid in *S. cerevisiae*. The authors should compare with those works and emphasize the novelty and significance of this study.

Some minor points:

1. What are the copy numbers of the genes integrated to the cell genome?
2. Have the authors tuned the expression of the inserted genes to optimize the metabolic flow?
3. In Line 204, references are needed.
4. In Lines 222-223, At what concentration, hexanoic acid significantly inhibit *Yarrowia* growth?
5. The retention times of the three samples are difficult. Did the authors perform MS to confirm?
6. How many biological repeats were performed?
7. The vendors of the chemicals are not mentioned.
8. In Figure 5, what are the ODs?

Reviewer #2 (Remarks to the Author):

This manuscript describes the demonstration of the microbial production of olivetolic acid, a precursor of neurologically active, medically valuable compounds cannabinoids, by overexpression of olivetolic acid synthase and olivetolic acid cyclase in oleaginous yeast *Yarrowia lipolytica*, and subsequent improvement of its titer by 83-fold through overexpression of critical enzymatic steps for supply of hexanoyl-CoA, malonyl-CoA, acetyl-CoA, NADPH and ATPs. This result could offer the basis for further enhancement of production of olivetolic acid and demonstration of productions of cannabinoids in oleaginous yeasts, which possess some superior properties to other common microbial hosts. The achievement shown in this manuscript is remarkable and the authors tried hard to address the importance of this result, but the logic and flow are still not very clear, and the manuscript seems to be not meticulously drafted. Therefore, problems and questions listed below need to be fixed and well addressed before final acceptance.

1. The reasons why *Yarrowia lipolytica* is chosen as the host compared to other common hosts are not described in a clear and logical flow. What is the constraint of common hosts *E. coli* and *S. cerevisiae*? How the properties *Yarrowia lipolytica* are advantageous for production of olivetolic acid and

cannabinoids? Why involvement of polyketide synthase, mevalonate pathway and prenyltransferase in cannabinoid synthesis poses a challenge for common hosts? What is the reference of the 'report' that *Yarrowia lipolytica* can accommodate strong flux of acetyl-CoA, malonyl-CoA and HMG-CoA? These questions need to be clearly answered to clearly illustrate the reasons why *Yarrowia lipolytica* is chosen. Furthermore, though the paper claims that the olivetolic acid titer in *Yarrowia lipolytica* is higher than the reported data in *S. cerevisiae*, but it is lower than the reported titer in *E. coli*. What is the reason and is there any clear solution to let *Yarrowia lipolytica* outperform *E. coli*?

2. What is the evidence that the supplies of hexanoyl-CoA, malonyl-CoA, acetyl-CoA, NADPH and ATPs are 'rate-limiting' for olivetolic acid production? Just because their overexpression can help improve the production of olivetolic acid? It would be better to provide more direct proofs to claim that they are 'rate-limiting'. Moreover, the manuscript lacks the description of impacts of NADPH supply on olivetolic acid production as olivetolic acid synthase and cyclase themselves don't need NADPH cofactors and the reason why malic enzyme is chosen for overexpression to improve the NADPH supply. In addition, it would be better to measure the intracellular levels of hexanoyl-CoA, malonyl-CoA, acetyl-CoA, NADPH and ATPs to prove that after the enzymatic operations, their levels are really increased.

3. Is there any spontaneous decarboxylation of olivetolic acid to olivetol detected during the production? What is the stability of olivetolic acid in different media used in this research? Does *Yarrowia lipolytica* consume olivetolic acid in these different media? Feeding of olivetolic acid standards in media with or without cells could help address these questions and might answer the difference of titers in different media.

4. Many data seem to not have significant differences, like titers from YL104 with or without feeding of hexanoic acid as shown in Fig. 3a, titers in different media shown in Fig. 5b, and titers in YL127 and YL128 as shown in Fig. 5c. It would be better to provide p-values to prove that these changes are significant, not within the error ranges. Moreover, please provide the number of biological replicates for each test.

5. The figures and their captions do not provide enough clear information. Please add the phenotypes or genotypes of each strain in the figure in a concise way. Also, not all engineered steps shown in Figure 1 are beneficial for olivetolic acid production like DGA1 and DGA2 deletions, so please point out the steps that improved the olivetolic acid production.

6. Why the effect of pH on olivetolic acid production is described in the same section of effect of malonyl-CoA supply in the main text? Please offer more detailed reason that pH is related with malonyl-CoA. Then, why the relevant figure, namely Figure 5a, is in displayed with the figures on effects of pathway integration, media, acetyl-CoA, ATP and NADPH, not with the figures about effects of malonyl-CoA supply in Figure 4?

7. In Lines 315-317, how it is speculated that acetyl-CoA supply is a bottleneck based on the results between pathway expressions from pathway or genomic integration? Also, what is reason of difference of titers when integrating in different locus?

8. Are there any efforts to further improve the production by optimizing the growth conditions like volume, temperature, inducer concentration and aeration level? Why scale-up in a fermenter has not been tried?

9. The words 'to improve acetyl-CoA supply' in Line 318 is redundant as there are words 'to boost the availability of acetyl-CoA' in the same sentence. Also, in section 2.6, the words 'in this work' and 'here' seem not appropriate. Is this section directly copied from somewhere else? Please re-paraphrase it. In addition, there are some other typos and other grammatical errors, so please re-scrutinize and revise the whole manuscript carefully.

10. Last but not least, CBD is the acronym of cannabidiol, a compound belonging to cannabinoid, not the whole cannabinoids. Please correct it to avoid any possible confusion.

Reviewer #3 (Remarks to the Author):

This article describes a metabolic engineering strategy to produce cannabinoid precursor in *Y.*

lipolytica. The work is descriptive and the results are interesting. Nonconventional yeasts are of growing interest for chemicals production and the elevated titers produced in this initial work are promising for cannabinoid production. Like many works of this nature, the pathway or product is not necessarily novel, but the host is changed, resulting in higher titers after overcoming engineering challenges with the new host. I think it is suitable for publication after minor revisions to the writing. Particularly, the minor phrasing and typos listed in the comments but also much of the results could be more concise. The paper is very chronological and much of the approaches that didn't work could be briefly mentioned and moved to a supplement. This would enhance readability and clarity.

Minor Comments

- What was the rationale for the retrosynthetic approach? Certainly since the pathway is known one could engineer in the other direction – i.e. keep overproducing precursors sequentially. Why retrosynthesis?
- Line 17 – consider rephrasing “fight against” it is a bit too symbolic
- Line 24 – Yarrowia misspelled
- Line 25 – “overcome the supply” doesn't make sense – maybe increase the supply?
- Line 27 – shake flask?
- Line 41 – et al. maybe should be etc? or revised out.
- Line 53 – the economics are given in two different units /kg and /mg. can these be placed on the same basis? Also what is the source for these numbers?
- Line 56 – pandemic
- Line 69 – parenthetical not clear. Should be “hosts” because two organisms are listed
- Line 72 – et al. should be etc. or revised out
- Line 198 – “the Cannabis species” which one? The sentence just ends.
- The paragraph beginning on line 205 is too long to discuss an approach that did not work.
- Line 223 – inhibits
- The paragraph beginning on line 217 is too long and should be revised.
- The paragraph beginning on line 239 doesn't explain why p. putida was sourced for the genes and also is hard to follow each engineering step since it covers several.

Dear Editors and Reviewers,

We apologize for the delay in preparing the revised manuscript. Due to the impact of politics and pandemics, our lab was relocated to Technion-Guangdong, Israel Institute of Technology last October. All the research materials, including strains and plasmids were kept in the US, University of Maryland Baltimore County.

We have thoroughly revised the manuscript. Here is a point-to-point response to address three reviewer's comments. We hope this may clear the concerns.

Thanks for offering the opportunity to have this article revised and critically commented. We highly appreciate your time to read this article.

Point-to-point response to reviewers' comments

Reviewer #1 (Remarks to the Author):

In this manuscript, the authors engineer an oleaginous yeast, *Yarrowia lipolytica*, to produce olivetolic acid, a key precursor of cannabinoid biosynthesis. The authors optimize the metabolic pathway, especially the biosynthesis of Hexanoyl-CoA and Malonyl-CoA. Although previous studies have successfully constructed *E. coli* and *S. cerevisiae* strains to produce olivetolic acid and cannabinoids, the authors argue that *Yarrowia* could be a promising host in large-scale cannabinoid production.

My major concern is that the high toxicity of hexanoic acid to *Yarrowia* would be the bottleneck of olivetolic acid production. Have the authors tried to synthesize Hexanoyl-CoA from Acetyl-CoA as Ref. 22 did?

Response: No, we did not try to synthesize hexanoyl-CoA from acetyl-CoA as Ref. 22 did. In Ref. 22, the authors engineered a reversed β -oxidation pathway in *S. cerevisiae* to produce hexanoyl-CoA by using acetyl-CoA as both the starter and extender units, but the titer of olivetolic acid produced in the resulting strain was lower than that in the engineered strain fed with hexanoic acid. Actually, the reversal of β -oxidation for chemical synthesis was first

demonstrated in *E. coli* under anaerobic or microaerobic conditions. The first report on the reversal of the β -oxidation cycle in *S. cerevisiae* was published in 2014 (Jiazhang Lian and Huimin Zhao, Reversal of the β -Oxidation Cycle in *Saccharomyces cerevisiae* for Production of Fuels and Chemicals, ACS Synth. Biol. 2015, 4, 332–341), but this system seemed to not work well in yeast when compared with the high efficiency of the reversed β -oxidation pathway in *E. coli*. The authors explained that the cellular acetyl-CoA level was the rate-limiting factor of the reversed β -oxidation pathway. Later, another article published and the authors concluded that “We conclude that expression of the reverse β -oxidation pathway in *S. cerevisiae* poses many challenges when compared to expression in bacterial systems. These factors gravely hinder development efforts and success rate for producing fatty acids through this pathway.” (<https://www.biorxiv.org/content/10.1101/201616v1.full>).

In the future work, we will try to construct a functional reversed β -oxidation pathway in *Yarrowia lipolytica*. Both *E. coli* and *S. cerevisiae* are facultative anaerobic, while *Y. lipolytica* is aerobic. Until now, there is no report on the reversed β -oxidation pathway in *Y. lipolytica*. Therefore, a lot of work needs to be done to construct an efficient reversed β -oxidation pathway in *Y. lipolytica*. We hope this explanation may be helpful to clear some of the concerns.

Previous studies, such as Ref. 22, had successfully produced olivetolic acid by overexpressing CsAAE1, CsOLS and CsOAC while feeding hexanoic acid in *S. cerevisiae*. The authors should compare with those works and emphasize the novelty and significance of this study.

Response: Thank you very much for the suggestion. We have added the sentence “The overexpression of *CsAAE1* in *S. cerevisiae* only showed a twofold increase in OLA titer (Ref.22). By contrast, *PpLvaE* performs best to convert hexanoic acid to hexanoyl-CoA in all the recent reports. This is the first report to use *PpLvaE* to overcome a rate-limiting step for constructing the OLA biosynthesis pathway.” and cited the corresponding citation into the section 3.2. In our paper, we identified the codon-optimized *PpLvaE* is the most efficient step to improve hexanoyl-CoA level in *Yarrowia*, this is the first report.

Some minor points:

1. What are the copy numbers of the genes integrated to the cell genome?

Response: The genes integrated to the cell genome are one copy. We are using the PBR322 docking platform, which is a built-in feature with 2kb homologues arm on both sides in our *Polg* and *Polf* strains. We have added the sentence “All the genes manipulated in this study are one copy.” at the end of section 2.6.

2. Have the authors tuned the expression of the inserted genes to optimize the metabolic flow?

Response: No, we have not tuned the expression of the inserted genes to optimize the metabolic flow. This will be our future focus.

3. In Line 204, references are needed.

Response: Reference 22 is cited in line 204.

4. In Lines 222-223, At what concentration, hexanoic acid significantly inhibit *Yarrowia* growth?

Response: As shown in Figure 3b and 3c, hexanoic acid can significantly inhibit *Yarrowia* growth at 0.5 mM or 1 mM when we fed *Yarrowia* with hexanoic acid at 24 h or 48 h, respectively.

5. The retention times of the three samples are difficult. Did the authors perform MS to confirm?

Response: We only perform MS to confirm olivetolic acid. A detailed UHPLC-MS report could be found in the supplementary information.

6. How many biological repeats were performed?

Response: All reported results are three biological replicates with standard deviations. We have added the sentence “Error bars represent standard deviation of three biological replicates.” in the figure legends.

7. The vendors of the chemicals are not mentioned.

Response: Fixed. In our revised manuscript, they are mentioned in section 2.1. Olivetolic acid was purchased from Santa Cruz Biotechnology. All other chemicals were obtained from Sigma-Aldrich and Fisher Scientific.

8. In Figure 5, what are the ODs?

Response: As shown in Figure 3c, the supplementation of 0.5 mM hexanoic acid at 48 hours was used for further strain screening. Thus, the OD₆₀₀ values of the strains in Figure 5 were around 25 at fermentation for 96 h.

Reviewer #2 (Remarks to the Author):

This manuscript describes the demonstration of the microbial production of olivetolic acid, a precursor of neurological active, medically valuable compounds cannabinoids, by overexpression of olivetolic acid synthase and olivetolic acid cyclase in oleaginous yeast *Yarrowia lipolytica*, and subsequent improvement of its titer by 83-fold through overexpression of critical enzymatic steps for supply of hexanoyl-CoA, malonyl-CoA, acetyl-CoA, NADPH and ATPs. This result could offer the basis for further enhancement of production of olivetolic acid and demonstration of productions of cannabinoids in oleaginous yeasts, which possess some superior properties to other common microbial hosts.

The achievement shown in this manuscript is remarkable and the authors tried hard to address the importance of this result, but the logic and flow are still not very clear, and the manuscript seems to be not meticulously drafted. Therefore, problems and questions listed below need to be fixed and well addressed before final acceptance.

Response: We thank the reviewers for making the comments. We tried our best to improve the logic flow of the revised manuscript. We thoroughly revised the manuscript to improve the clarity and readability.

1. The reasons why *Yarrowia lipolytica* is chosen as the host compared to other common hosts are not described in a clear and logical flow. What is the constraint of common hosts E.

coli and *S. cerevisiae*? How the properties *Yarrowia lipolytica* are advantageous for production of olivetolic acid and cannabinoids? Why involvement of polyketide synthase, mevalonate pathway and prenyltransferase in cannabinoid synthesis poses a challenge for common hosts? What is the reference of the ‘report’ that *Yarrowia lipolytica* can accommodate strong flux of acetyl-CoA, malonyl-CoA and HMG-CoA? These questions need to be clearly answered to clearly illustrate the reasons why *Yarrowia lipolytica* is chosen. Furthermore, though the paper claims that the olivetolic acid titer in *Yarrowia lipolytica* is higher than the reported data in *S. cerevisiae*, but it is lower than the reported titer in *E. coli*. What is the reason and is there any clear solution to let *Yarrowia lipolytica* outperform *E. coli*?

Response: Thank you for making these comments. For heterologous natural product pathway reconstitution, *E. coli* lacks eukaryotic subcellular organelles for expression of plant enzymes, In general, post-translational modifications and spatial localization were needed for expression of plant enzymes. The spatial sequestration of certain biosynthetic precursors and co-factors were also different in *E. coli* and yeast. *S. cerevisiae* has limited internal membrane structures compared to oleaginous yeast, and has different post-translational mechanisms compared to plants [1]. For commercial applications, *E. coli* has drawbacks such as bacterial toxin contamination, acetate inhibition, low activity of expressed plant-based enzymes, and an inability to post-translationally modify and compartmentalize complex proteins. *S. cerevisiae* also has drawbacks such as relatively slow growth rates, complex growth medium requirements, and high ethanol production known as the Crabtree effect [2-4].

Olivetolic acid and cannabinoids synthesis involves polyketide synthases, mevalonate pathway and prenyltransferase, which use acetyl-CoA, malonyl-CoA and HMG-CoA as precursors. *E. coli* natively lacks MVA pathway and has low activity of expressed plant-based enzymes. *S. cerevisiae* has very limited amount of cytosolic acetyl-CoAs [5 and Reference 41]. As *Yarrowia lipolytica* is an oleaginous yeast capable of synthesizing large amount of lipids, naturally, they can accommodate strong flux of acetyl-CoA, malonyl-CoA, which are precursors to synthesize fatty acids and lipids. Recent studies also report *Yarrowia* is a superior host to synthesize squalene, which is directly derived from HMG-CoA. In this sense, *Y. lipolytica* is a promising host to produce olivetolic acid and cannabinoids. *Yarrowia*

lipolytica contains the native mevalonate (MVA) pathway to provide precursor compounds, namely isopentenyl pyrophosphate and dimethylallyl pyrophosphate, indicating that *Y. lipolytica* can serve as a natural host for terpenoid synthesis. The lipogeneity of this yeast makes it a superior host to produce chemicals that are derived from acetyl-CoA, malonyl-CoA, HMG-CoA [Reference 13].

Polyketide-producing organisms are typically plants or microbes that are not well-suited for high-level industrial production. In this work, we explored the possibility whether *Yarrowia lipolytica* can be a good platform to synthesize CBD precursors. When using *E. coli* as the host to produce olivetolic acid (OLA), the authors optimized malonyl-CoA and hexanoyl-CoA supply and then only obtained 0.71 mg/L and 0.8 mg/L OLA in *E. coli* BL21 (DE3) and K-12 MG1655 (DE3) strains, respectively. However, the OLA production increased greatly when the author sought to exploit a previously engineered *E. coli* JST10(DE3) strain. This strain not only has β -oxidation reversal (r-BOX) pathway for generating hexanoyl-CoA, but also has fermentative product pathways (e.g., lactate, succinate, acetate and ethanol) and thioesterases (e.g., *tesA* and *tesB* among others) deleted to ensure adequate acetyl-CoA supply and minimize the loss of acyl-CoA intermediates.

In the future, there are some solutions that may let *Y. lipolytica* outperform *E. coli*. The first strategy is to further increase acetyl-CoA supply and to delete acetyl-CoA competitive and consumption pathways. The second strategy is protein engineering strategies or directed evolution of *PpLvaE* that may further improve hexanoyl-CoA supply. The third strategy is that additional copies of genes can be introduced into the yeast for overexpression. The fourth strategy is to improve the bioprocess parameters by optimizing the growth conditions like volume, pH, temperature, inducer concentration and aeration level or using biphasic fermentation with solvent overlay *et al.*

References:

- [1] Carqueijeiro I, Langley C, et al. Beyond the semi-synthetic artemisinin: metabolic engineering of plant-derived anti-cancer drugs. *Current Opinion in Biotechnology* 2019, 65: 17-24.
- [2] Wang C, Liwei M, Park J-B, Jeong S-H, Wei G, Wang Y, Kim S-W. Microbial platform for terpenoid production: *Escherichia coli* and yeast. *Front Microbiol* 2018, 9:2460.

[3] Moser S, Pichler H. Identifying and engineering the ideal microbial terpenoid production host. *Appl Microbiol Biotechnol* 2019, 103:5501-5516.

[4] Immethun C, Hoynes-O'Connor A, Balassy A, Moon TS. Microbial production of isoprenoids enabled by synthetic biology. *Front Microbiol* 2013, 4:75.

[5] Chen Y, Daviet L, Schalk M, Siewers V, Nielsen J. Establishing a platform cell factory through engineering of yeast acetyl-CoA metabolism. *Metab Eng.* 2013; 15:48–54.

[6] Ahmad M Abdel-Mawgoud, Kelly A Markham, Claire M Palmer, Nian Liu, Gregory Stephanopoulos, Hal S Alper. Metabolic engineering in the host *Yarrowia lipolytica*. *Metabolic engineering* 2018, 50:192-208.

[7] Kelly A Markham, Hal S Alper. Synthetic biology expands the industrial potential of *Yarrowia lipolytica*. *Trends in biotechnology* 2018, 36: 1085-1095.

[8] Yufen Wu, Shuo Xu, Xiao Gao, Man Li, Dashuai Li & Wenyu Lu. Enhanced protopanaxadiol production from xylose by engineered *Yarrowia lipolytica*. *Microbial Cell Factories* 2019, 18:83.

2. What is the evidence that the supplies of hexanoyl-CoA, malonyl-CoA, acetyl-CoA, NADPH and ATPs are ‘rate-limiting’ for olivetolic acid production? Just because their overexpression can help improve the production of olivetolic acid? It would be better to provide more direct proofs to claim that they are ‘rate-limiting’. Moreover, the manuscript lacks the description of impacts of NADPH supply on olivetolic acid production as olivetolic acid synthase and cyclase themselves don’t need NADPH cofactors and the reason why malic enzyme is chosen for overexpression to improve the NADPH supply. In addition, it would be better to measure the intracellular levels of hexanoyl-CoA, malonyl-CoA, acetyl-CoA, NADPH and ATPs to prove that after the enzymatic operations, their levels are really increased.

Response: We thank the reviewers to make such important comments. Due to the impact of politics and pandemics, our lab has relocated to China, all the research materials, including strains and plasmids were kept in the US, University of Maryland Baltimore County. We apologize that we couldn’t retrieve the strains to quantitatively analyze the hexanoyl-CoA, malonyl-CoA, acetyl-CoA, NADPH and ATPs level in our study. We do believe a thorough

metabolomic analysis will be the key to troubleshoot and remove the pathway bottlenecks in future studies.

Even without direct experimental evidence, literature reports have confirmed our assumptions. For example, a report (Ref 21) with *in vitro* analysis has confirmed the potential for CsOLS and CsOAC expressed in *E. coli* to synthesize OLA from hexanoyl-CoA and malonyl-CoA. Codon optimized, His-tagged CsOLS and CsOAC were expressed and purified from *E. coli* and utilized to determine product formation in a reaction system including hexanoyl-CoA (primer) and malonyl-CoA (extender unit). The incubation of CsOLS and CsOAC in the presence of these substrates resulted in OLA synthesis. To evaluate their ability to produce OLA *in vivo*, the plasmid containing codon-optimized versions of CsOLS and CsOAC was transformed into *E. coli* BL21. However, the OLA titer was very low (~0.1 mg/L). In reference 22, when the authors introduced a CsOLS and CsOAC expression cassette into *S. cerevisiae*, the engineered strain produced 0.2 mg/L OLA. Therefore, in our work, the plasmid encoding the codon-optimized CsOLS and CsOAC was transformed into *Y. lipolytica*, the resulting strain produced 0.11 mg/L OLA. The OLA titers were very low in *E. coli*, *S. cerevisiae* and *Y. lipolytica* when expressing CsOLS and CsOAC *in vivo*. It could be speculated that a major limitation for OLA production is the availability of required precursors, rather than issues with the expression or activity of these enzymes. In reference 21, by combining pathways with auxiliary enzymes for additional hexanoyl-CoA and malonyl-CoA generation, the authors identified the supply of these precursors as a key limiting factor in OLA synthesis. In our current work, similarly, we found that increasing the supplies of hexanoyl-CoA, malonyl-CoA, acetyl-CoA, NADPH and ATP can improve the production of OLA, so precursor supply is a major limiting factor for OLA synthesis.

NADPH, the primary biological reducing equivalent to protect cell from oxidative stress and extend carbon-carbon backbones, has been reported as the major rate-limiting precursor in fatty acids and lipids synthesis in oleaginous species (references 32, 38 and 44). β -oxidation cycle can provide cytosolic acetyl-CoA from fatty acids. In reference 32, the PI's lab has improved the production of the polyketide triacetic acid lactone (TAL) in *Y. lipolytica* by recycling cytosolic NADPH. All the four alternative cytosolic NADPH pathways can improve TAL production in *Y. lipolytica*. And malic enzyme was found as the best target.

Thus, in this study, NADPH was supplied for the polyketide OLA production. In the manuscript, the sentence in lines 300-302 is revised to “and the biosynthesis of OLA from acetyl-CoA requires extensive consumption of ATP and NADPH as cofactors to facilitate the activity of ACC, hexanoyl-CoA synthetase and the synthesis of fatty acids in oleaginous species^{21, 32, 38}”.

In reference 32, malic enzyme presented the best results to improve TAL production among the four chosen NADPH source pathways, despite that some report mentioned that malic enzyme in *Yarrowia lipolytica* might be localized into the mitochondria. Thus, in this study, malic enzyme is also chosen for overexpression to reconfirm its ability to improve cytosolic NADPH supply. In addition, pyruvate can be produced through the reaction catalyzed by malic enzyme, and can be used as substrate to generate cytosolic acetyl-CoA by *E. coli* pyruvate dehydrogenase complex (*EcPDH*) with the lipoate-protein ligase A (*EcLplA*) (references 32). Thus, in line 333, we added the sentence “Also, pyruvate can be produced through the reaction catalyzed by malic enzyme and can be converted to acetyl-CoA by *EcPDH* with *EcLplA* which could synergistically boost cytosolic acetyl-CoA in *Y. lipolytica*³².”

Due to relocation of our lab from USA to China, we did not measure the intracellular levels of hexanoyl-CoA, malonyl-CoA, acetyl-CoA, NADPH and ATPs. All research materials, including strains and plasmids, were locked at University of Maryland. We hope this may help clear some of the concerns raised by the reviewers.

3. Is there any spontaneous decarboxylation of olivetolic acid to olivetol detected during the production? What is the stability of olivetolic acid in different media used in this research? Does *Yarrowia lipolytica* consume olivetolic acid in these different media? Feeding of olivetolic acid standards in media with or without cells could help address these questions and might answer the difference of titers in different media.

Response: We would like to thank the reviewer for making this important comment. Previous studies indicate that olivetol can be detected during olivetolic acid production (Ref. 21). The authors found that OLA levels decreased over time, with the majority of OLA decarboxylated to olivetol. Due to the impact of politics and pandemics, our lab was relocated to

Technion-Guangdong, Israel Institute of Technology last October. All the research materials, including strains and plasmids were kept in the US, University of Maryland Baltimore County. We couldn't test how the engineered yeast will metabolize olivetolic acids. It is likely that OA is not stable and will be decarboxylated into olivetol in *Yarrowia*. In our new lab, we are investigating the stability of OLA and whether *Yarrowia lipolytica* consumes OLA in different media, we are also going to identify the enzyme or reactions that will decarboxylate or oxidize OAs. These studies will be critical for us to pull the OA flux towards downstream CBDs.

4. Many data seem to not have significant differences, like titers from YL104 with or without feeding of hexanoic acid as shown in Fig. 3a, titers in different media shown in Fig. 5b, and titers in YL127 and YL128 as shown in Fig. 5c. It would be better to provide p-values to prove that these changes are significant, not within the error ranges. Moreover, please provide the number of biological replicates for each test.

Response: Thank you very much for the suggestion. The two-tailed *t*-test method is employed to analyze the statistical significance of the data in Fig. 3a, Fig. 5b and Fig. 5c, and *p*-value < 0.05 is deemed statistically significant. All reported results in this study are three biological replicates with standard deviations. We have added the sentence “The asterisk indicates the statistical significance at a *p* < 0.05 (two-tailed *t*-test).” and “Error bars represent standard deviation of three biological replicates.” in the figure legends.

5. The figures and their captions do not provide enough clear information. Please add the phenotypes or genotypes of each strain in the figure in a concise way. Also, not all engineered steps shown in Figure 1 are beneficial for olivetolic acid production like DGA1 and DGA2 deletions, so please point out the steps that improved the olivetolic acid production.

Response: Thank you very much for the suggestion. We have added the genotypes of each strain in the figures. The engineered steps that improved the olivetolic acid production are colored blue in Figure 1. We have added the sentence “Blue colored steps are beneficial for olivetolic acid production” in the figure legend.

6. Why the effect of pH on olivetolic acid production is described in the same section of effect of malonyl-CoA supply in the main text? Please offer more detailed reason that pH is related with malonyl-CoA. Then, why the relevant figure, namely Figure 5a, is in displayed with the figures on effects of pathway integration, media, acetyl-CoA, ATP and NADPH, not with the figures about effects of malonyl-CoA supply in Figure 4?

Response: Thanks for making this comment. In our revised manuscript, we have rearranged our manuscript structure and place the effect of pH on olivetolic acid production into a separate section 3.4 in our revised manuscript. We hope this may help improve the clarity of the reported work.

7. In Lines 315-317, how it is speculated that acetyl-CoA supply is a bottleneck based on the results between pathway expressions from pathway or genomic integration? Also, what is reason of difference of titers when integrating in different locus?

Response: After the integration of the plasmid *pURLB-CsOLS-CsOAC-PpLvaE-ylACCI* at different genomic locus, the OLA titer was not further improved. The supplies of hexanoyl-CoA and malonyl-CoA were already optimized by the expressing the genes *PpLvaE* and *ylACCI*. Given that acetyl-CoA served as the basic building block for both hexanoyl-CoA and malonyl-CoA, we reasoned that acetyl-CoA supply is a bottleneck.

Gene expression via genomic integration are regulated and impacted by multifaceted factors in yeast. For examples, In the reference 39, the authors concluded that gene expression in eukaryotes is a complex process regulated at multiple levels, and transcription of transgenes is not only influenced by recombinant promoters and regulatory DNA elements but also by the spatial positioning of the transgenes within the genome. It indicates that the position effect of integration loci of genome on heterologous gene expression should be considered. It had been reported that the position effect of integration loci existed in *E. coli*, *S. cerevisiae*, *Lactococcus lactis*, *Drosophila* and human genome [1-6]. Some researchers have characterized 20 different integration sites of the *S. cerevisiae* genome by inserting *lacZ* as a reporter gene and determining expression levels. An up to 8.7-fold difference was detected between the sites conferring lowest and highest expression, respectively [3]. We hope this explanation might be helpful to clear the concerns raised by the reviewers.

References:

- [1] Sousa C, de Lorenzo V, Cebolla A. Modulation of gene expression through chromosomal positioning in *Escherichia coli*. *Microbiology*, 1997, 143 (6): 2071–2078.
- [2] Yamane S, Yamaoka M, Yamamoto M, et al. Region specificity of chromosome III on gene expression in the yeast *Saccharomyces cerevisiae*. *J Gen Appl Microbiol*, 1998, 44 (4): 275 – 281.
- [3] Bai Flagfeldt D, Siewers V, Huang L, et al. Characterization of chromosomal integration sites for heterologous gene expression in *Saccharomyces cerevisiae*. *Yeast*, 2009, 26 (10): 545–551.
- [4] Siezen RJ, Bayjanov JR, Felis GE, et al. Genome-scale diversity and niche adaptation analysis of *Lactococcus lactis* by comparative genome hybridization using multi-strain arrays. *Microb Biotechnol*, 2011, 4 (3): 383–402.
- [5] Markstein M, Pitsouli C, Villalta C, et al. Exploiting position effects and the gypsy retrovirus insulator to engineer precisely expressed transgenes. *Nat Genet*, 2008, 40 (4): 476–483.
- [6] Gierman HJ, Indemans MHG, Koster J, et al. Domain-wide regulation of gene expression in the human genome. *Genome Res*, 2007, 17 (9): 1286–1295.

8. Are there any efforts to further improve the production by optimizing the growth conditions like volume, temperature, inducer concentration and aeration level? Why scale-up in a fermenter has not been tried?

Response: Thank you for the comments. We did not further improve the production by optimizing the growth conditions. This work is the effort of two PhD scientists working two years from Spring 2018 to Spring 2020. Due to the impact of politics and pandemics, my lab was relocated to Technion-Guangdong, Israel Institute of Technology. All research materials, including strains and plasmids, were locked at University of Maryland. For this reason, we didn't perform bioreactor-based experiments in this work. We will reconstruct the strains, further improve the production by optimizing the growth conditions in the future work when conditions are ready.

9. The words ‘to improve acetyl-CoA supply’ in Line 318 is redundant as there are words ‘to boost the availability of acetyl-CoA’ in the same sentence. Also, in section 2.6, the words ‘in this work’ and ‘here’ seem not appropriate. Is this section directly copied from somewhere else? Please re-paraphrase it. In addition, there are some other typos and other grammatical errors, so please re-scrutinize and revise the whole manuscript carefully.

Response: Thank you for the comments. The words ‘to improve acetyl-CoA supply’ in line 318 and ‘here’ in line 161 are deleted. We also thoroughly revised other typos and grammatical errors in the manuscript.

10. Last but not least, CBD is the acronym of Cannabidiol, a compound belonging to cannabinoid, not the whole cannabinoids. Please correct it to avoid any possible confusion.

Response: Thanks. We fixed the meaning of ‘CBD’ in the manuscript.

Reviewer #3 (Remarks to the Author):

This article describes a metabolic engineering strategy to produce cannabinoid precursor in *Y. lipolytica*. The work is descriptive and the results are interesting. Nonconventional yeasts are of growing interest for chemicals production and the elevated titers produced in this initial work are promising for cannabinoid production. Like many works of this nature, the pathway or product is not necessarily novel, but the host is changed, resulting in higher titers after overcoming engineering challenges with the new host. I think it is suitable for publication after minor revisions to the writing. Particularly, the minor phrasing and typos listed in the comments but also much of the results could be more concise. The paper is very chronological and much of the approaches that didn’t work could be briefly mentioned and moved to a supplement. This would enhance readability and clarity.

Response: Thanks for the comments. We have fully revised the manuscript to enhance its readability and clarity.

Minor Comments

- What was the rationale for the retrosynthetic approach? Certainly since the pathway is known one could engineer in the other direction – i.e. keep overproducing precursors

sequentially. Why retrosynthesis?

Response: Retrosynthetic approach is achieved by dissecting a target biomolecule into simpler precursor molecules that have a biological origin. This procedure is repeated until all known metabolic pathways were clear for each of the precursor molecules. Then from these precursor molecules, we will be able to reconstitute each section of the pathway and synthesize the final compounds. Often, a target compound will have more than one possible synthetic route. Retrosynthesis is well suited for discovering different synthetic routes and comparing them in a logical and straightforward manner. For example, in this study, malonyl-CoA can be synthesized from two synthetic routes since malonyl-CoA can be broken into malonate and coenzyme A or acetyl-CoA and bicarbonate (HCO_3^-). We chose acetyl-CoA as the precursor to synthesize malonyl-CoA, because *Y. lipolytica* can accommodate strong flux of acetyl-CoA. Another advantage of taking a retrosynthetic step is that we can sequentially troubleshoot the pathway bottlenecks in a reversed manner. For example, from our target molecules, we can deduce rate-limiting steps by feeding relevant precursor molecules to the cell expressing the relevant enzymes. Consumption of the precursor molecules or the accumulation of the final products will bear critical information for us to optimize the reaction rates. We hope this explanation may help clear some of the concerns.

- Line 17 – consider rephrasing “fight against” it is a bit too symbolic

Response: We revised the words “fight against” to “treat”.

- Line 24 – Yarrowia misspelled

Response: We revised the word “Yarrowai” to “Yarrowia”.

- Line 25 – “overcome the supply” doesn’t make sense – maybe increase the supply?

Response: We revised the word “overcome” to “increase”.

- Line 27 – shake flask?

Response: We revised the word “shaking” to “shake”.

- Line 41 – et al. maybe should be etc? or revised out.

Response: We revised “et al.” to “etc.”.

- Line 53 – the economics are given in two different units /kg and /mg. can these be placed on the same basis? Also what is the source for these numbers?

Response: Fixed. The revised text now reads “The overall market size of CBD oil as consumer chemicals is estimated to be \$25 billion in 2025 (assuming 50 Million people will use CBDs as a consumer chemical daily, average dosage 50g/people/year or 2 mg/kg-body-weight/day). Current crude CBD oil (45% strength) is sold at about \$3,000/kg and pure CBD oil is sold at \$10,000/kg.”

- Line 56 – pandemic

Response: We revised the word “pandemics” to “pandemic”.

- Line 69 – parenthetical not clear. Should be “hosts” because two organisms are listed

Response: We revised the word “host” to “hosts”.

- Line 72 – et al. should be etc. or revised out

Response: We revised “et al.” to “etc.”.

- Line 198 – “the Cannabis species” which one? The sentence just ends.

Response: We added “(*C. sativa*)” behind “the Cannabis species”.

- The paragraph beginning on line 205 is too long to discuss an approach that did not work.

Response: In our revised manuscript, the paragraph has been simplified. Now the text reads “Evidence showed that some undesired byproducts could also be formed during OLA accumulation^{18, 19}. We hypothesized that fusing *CsOLS* with *CsOAC* may place the two enzymes in close proximity and minimize the dissipation of intermediates. In such a way, fused OLS-OAC may efficiently transfer the intermediates to form OLA with minimal byproducts. We fused the two proteins with an amino acid linker (10×Glycine) in two

different orientations. When *CsOLS* was fused to the N-terminus or C-terminus of the *CsOAC*, the obtained strains YL102 and YL103 showed a slightly decline in the production of OLA compared with the control strains YL101 (Figure 2c). Possibly, the fusion of these two proteins negatively impacted protein folding resulting in suboptimal catalytic functions. Since *CsOLS-CsOAC* fusion could not further improve OLA, the best production strain, YL101, was subjected to further engineering.”

- Line 223 – inhibits

Response: We revised the word “inhibit” to “inhibits”.

- The paragraph beginning on line 217 is too long and should be revised.

Response: Thanks for pointing this out. We simplified the entire paragraph. The new text reads “Pool size of hexanoyl-CoA was proven to be a rate-limiting factor in both *E. coli* and *S. cerevisiae*^{18,19}. An acyl activating enzyme encoded by *CsAAE1* from *Cannabis sativa* was characterized to catalyze the formation of hexanoyl-CoA from hexanoic acid, and the expression of *CsAAE1* has been shown to successfully increase the titer of OLA by twofold in *S. cerevisiae*¹⁹. Hexanoic acid is toxic and the dosage of hexanoic acid (HA) should be optimized to minimize its negative impact on cell fitness¹⁸. Thus, *CsAAE1* was co-expressed with *CsOLS* and *CsOAC* in *Y. lipolytica* supplemented with 0.5 mM, 1 mM, and 2 mM of HA. We found that OLA production was declined (Figure 3a). The yeast grows slowly with HA supplementation, confirming the negative effect of HA on cell growth (Figure 3b). To mitigate this effect, we delayed HA supplementation to 48 h. OLA production with 0.5 mM HA supplementation slightly increased to 0.13 mg/L (Figure 3a), and the yeast growth fitness remains unchanged (Figure 3c). Thus, the supplementation of 0.5 mM hexanoic acid at 48 hours was used for further strain screening.”.

- The paragraph beginning on line 239 doesn’t explain why *p. putida* was sourced for the genes and also is hard to follow each engineering step since it covers several.

Response: Revised and the next text reads “*Pseudomona sp*, known as a superior

environmental degrader for various carbon sources. The *LvaE* gene encoding the short chain acyl-CoA synthetase of *Pseudomonas putida* KT2440 (*PpLvaE*, PP_2795 with uniprot ID Q88J54), which is reported to active with C4-C6 carboxylic acids, was also selected²¹. Among these chosen hexanoyl-CoA synthetases, we found the overexpression of *PpLvaE* with *CsOLS* and *CsOAC* (strain YL110) resulted in the highest OLA production, an eightfold increase in OLA titer (1.07 mg/L) (Figure 4a). *P. putida* has been reported to exhibit better solvent tolerance and grow under a number of harsh or hydrophobic conditions. The overexpression of *CsAAE1* in *S. cerevisiae* only showed a twofold increase in OLA titer²². By contrast, *PpLvaE* outperforms other candidate enzymes in converting HA to hexanoyl-CoA. This is the first report that harnessed *PpLvaE* to overcome a rate-limiting step in OLA and cannabinoids pathway. Future enzymology study or directed evolution of *PpLvaE* may improve its catalytic function.”

REVIEWERS' COMMENTS:

Reviewer #1 (Remarks to the Author):

I recommend the acceptance of this manuscript.

Reviewer #2 (Remarks to the Author):

The authors have addressed some of the reviewer's comments, whereas some other comments, which were critical for determining the significance of this manuscript, have been ignored.

1. As reviewer #1, the toxicity of hexanoic acid was obviously the largest "rate-limiting step" needed to be solved, however, the authors did not respond to this question. At least the authors need to provide a possible reason and a tentative solution for this question, otherwise it is difficult to believe that *Y. lipolytica* could be a suitable host for cannabinoid production.

2. As in reviewer #2's comments, NADPH usually present as a rate-limiting step only when the product titer reached sub-gram or gram scale, as in those cited papers. However, in this manuscript, when adding MAE, the titer of OLA is lower than 10 mg/L, in combination of the fact that these enzymes also generate extra pyruvate/acetyl-CoA, it is risky to conclude that NADPH was the bottleneck.

Reviewer #3 (Remarks to the Author):

In this revision, the authors have addressed all but one minor typographical error that I noticed, and I do not have any further comments. I think the manuscript is suitable for publication.

- line 88 - "oleogeneity nature" please revise

REVIEWERS' COMMENTS:

Reviewer #1 (Remarks to the Author):

I recommend the acceptance of this manuscript.

>> We thank the reviewer for supporting this work.

Reviewer #2 (Remarks to the Author):

The authors have addressed some of the reviewer's comments, whereas some other comments, which were critical for determining the significance of this manuscript, have been ignored.

1. As reviewer #1, the toxicity of hexanoic acid was obviously the largest "rate-limiting step" needed to be solved, however, the authors did not respond to this question. At least the authors need to provide a possible reason and a tentative solution for this question, otherwise it is difficult to believe that *Y. lipolytica* could be a suitable host for cannabinoid production.

>> Thank you for making this positive comment. The toxicity is indeed the largest bottleneck to improve OA production. Short-chain fatty acids were reported to disrupt cell membrane integrity. While we cannot solve this challenge at this point, we do believe that *Y. lipolytica* will be a promising host compared to *E. coli* and *S. cerevisiae*. We have added this sentence in the discussion part "Apparently hexanoic acid toxicity poses severe inhibition to cell growth, and the hexanoyl-CoA ligase needs to be further improved. An evolutionary approach to improve hexanoic acid tolerance and selection of variants of hexanoyl-CoA ligase will be important to solve this challenge. Alternative approach is to harness the endogenous β -oxidation pathway and target OA pathway into the peroxisome."

2. As in reviewer #2's comments, NADPH usually present as a rate-limiting step only when the product titer reached sub-gram or gram scale, as in those cited papers. However, in this manuscript, when adding MAE, the titer of OLA is lower than 10 mg/L, in combination of the fact that these enzymes also generate extra pyruvate/acetyl-CoA, it is risky to conclude that NADPH was the bottleneck.

>> We agree with the reviewer. We have reworded our sentence in the revised manuscript.

Reviewer #3 (Remarks to the Author):

In this revision, the authors have addressed all but one minor typographical error that I noticed, and I do not have any further comments. I think the manuscript is suitable for publication.

>> We thank the reviewer for making the positive comments.

- line 88 - "oleogeneity nature" please revise

>> fixed, "oleogeneity nature" changed to "As an oleaginous yeast"